# Circulating cell free DNA during definitive chemo-radiotherapy in non-small cell lung cancer patients—Initial observations

Lotte Nygård[1]*, Lise B. Ahlborn[2], Gitte F. Persson[1], Dineika Chandrananda[3], Jonathan W. Langer[4], Barbara M. Fischer[4,5], Seppo W. Langer[1], Miglė Gabrielaite[2], Andreas Kjær[4], Nitzan Rosenfeld[3], Florent Mouliere[6], Olga Østrup[2], Ivan R. Vogelius[1], Søren M. Bentzen[7]

1 Department of Oncology, Rigshospitalet, Copenhagen University Hospital, Copenhagen, Denmark,
2 Center for Genomic Medicine, Rigshospitalet, Copenhagen University Hospital, Copenhagen, Denmark,
3 Cancer Research UK Cambridge Institute, University of Cambridge, Cambridge, England, United Kingdom,
4 Department of Clinical Physiology, Nuclear Medicine & PET and Cluster for Molecular Imaging, Rigshospitalet and University of Copenhagen, Copenhagen, Denmark, 5 PET Centre, School of Biomedical Engineering and Imaging Sciences, Kings College London, St Thomas' Hospital, London, England, United Kingdom, 6 Department of Pathology, Cancer Centre Amsterdam, Amsterdam UMC, Vrije Universiteit Amsterdam, Amsterdam, The Netherlands, 7 Division of Biostatistics and Bioinformatics, Department of Epidemiology and Public Health, University of Maryland Greenebaum Comprehensive Cancer Center, and University of Maryland School of Medicine, Baltimore, MD, United States of America

* lotte.nygaard@regionh.dk

**Data Availability Statement:** All relevant data are within the manuscript and its Supporting Information files.

## Abstract

### Background

The overall aim was to investigate the change over time in circulating cell free DNA (cfDNA) in patients with locally advanced non-small cell lung cancer (NSCLC) undergoing concurrent chemo-radiotherapy. Furthermore, to assess the possibility of detecting circulating cell free tumor DNA (ctDNA) using shallow whole genome sequencing (sWGS) and size selection.

### Methods

Ten patients were included in a two-phase study. The first four patients had blood samples taken prior to a radiation therapy (RT) dose fraction and at 30 minutes, 1 hour and 2 hours after RT to estimate the short-term dynamics of cfDNA concentration after irradiation. The remaining six patients had one blood sample taken on six treatment days 30 minutes post treatment to measure cfDNA levels. Presence of ctDNA as indicated by chromosomal aberrations was investigated using sWGS. The sensitivity of this method was further enhanced using *in silico* size selection.

### Results

cfDNA concentration from baseline to 120 min after therapy was stable within 95% tolerance limits of +/- 2 ng/ml cfDNA. Changes in cfDNA were observed during therapy with an apparent qualitative difference between adenocarcinoma (average increase of 0.69 ng/ml) and squamous cell carcinoma (average increase of 4.0 ng/ml). Tumor shrinkage on daily cone

**Funding:** SMB: This work was supported by the Danish Cancer Society grant no. R72-A4605-13-S2 and National Institutes of Health (NIH) grant no. P30 CA 134274-04. The authors further acknowledge financial support from the Kirsten and Freddy Johansen Foundation and Jubilæums Fonden at Rigshospitalet. https://www.cancer.dk/ The funders had no role in study design, data collection and analysis, decision to publish, or preparation of the manuscript.

**Competing interests:** The authors have declared that no competing interests exist.

beam computer tomography scans during radiotherapy did not correlate with changes in concentration of cfDNA.

## Conclusion

Concentrations of cfDNA remain stable during the first 2 hours after an RT fraction. However, based on the sWGS profiles, ctDNA represented only a minor fraction of cfDNA in this group of patients. The detection sensitivity of genomic alterations in ctDNA strongly increases by applying size selection.

## Background

Despite advances in curatively intended concurrent chemo-radiotherapy (cCRT), patients with inoperable NSCLC have a substantial risk of local or distant failure. At present, there are no liquid biomarkers available to predict treatment response or risk of relapse before or during cCRT in locally advanced, inoperable NSCLC [1],[2]. A recent report by the American Society of Clinical Oncology (ASCO) and College of American Pathologists does not recommend clinical implementation of any proposed biomarker due to little, if any evidence of clinical validity and utility in early stage cancers [3].

Circulating cell free DNA (cfDNA) are small fragments of DNA circulating in the blood stream and other body fluids. The mechanisms behind DNA release are still debated [4],[5]. It has been proposed that cfDNA originates from necrotic or apoptotic cells or by active release mechanisms [6],[7]. Short cfDNA fragments can be detected in plasma from cancer patients [8],[9] and plasma cfDNA concentrations from healthy individuals are shown to be lower than in lung cancer patients[10]. cfDNA in patients with disseminated NSCLC has been studied as a prognostic marker [11], and, a prospective study of 44 patients treated with chemotherapy could detect a correlation between baseline cfDNA and progression free survival (PFS) [12]. In addition, the detection of chromosomal instability quantification in plasma cfDNA has been associated with therapeutic response to immune therapy[13]. However, the use for total cfDNA measurements is still being investigated. Studies of the plasma levels of cfDNA in NSCLC patients treated with radiotherapy are sparse [14] and there is an unmet need of evaluating cfDNA analysis in this patient group. Furthermore, the half-life of cfDNA(clearance of DNA from the blood stream)and thus the adequate timing for blood sample collection remain unclear [4].

The present study is a prospective two-phase pilot study aiming at: 1) Determination of the most suitable time point for cfDNA blood sampling after chemo/radiotherapy and, 2) Assessment of changes in cfDNA in patients undergoing cCRT for evaluation of the feasibility of cfDNA as a potential biological response marker during cCRT. Analysis of ctDNA content in total cfDNA by shallow whole genome sequencing (sWGS) and in silico size selection was added for a subset of patients at time-points determined after analysis of cfDNA content.

## Methods

Consecutive patients with inoperable locally advanced NSCLC eligible for cCRT were offered inclusion at their first visit to the Department of Oncology before starting chemotherapy. Patients were offered 24 hours of consideration and could give their acceptance just before first cycle of chemotherapy. Patients were enrolled in the study from October 2016 until

September 2017 at the Department of Oncology, University Hospital of Copenhagen, Rigshospitalet, Denmark. Inclusion criteria were diagnosis of NSCLC, either adenocarcinoma (AC) or squamous cell carcinoma (SCC), age above 18 years and informed consent. Exclusion criteria were other active malignancies, antineoplastic treatment other than the scheduled chemo/radiotherapy for NSCLC, blood transfusion within 3 months of enrolment or, any active chronic inflammatory diseases e.g. rheumatoid arthritis. Chronic obstructive pulmonary disease was not considered an exclusion criterion.

All patients signed an informed consent prior to inclusion. The study was approved by the Danish Ethics Committee, protocol number H-16029201 and the Danish Data Protection Agency, 2012-58-0004/RH-2016-237/ I-suite no 04837 and by the institutional review board.

Treatment consisted of three cycles of cisplatin or carboplatin in combination with vinorelbine concurrent with radiotherapy delivered in 33 2-Gy fractions, five fractions per week to a total of 66 Gy. Cycles of chemotherapy were given three weeks apart. Radiotherapy started at the second cycle of chemotherapy and lasted 6.5 weeks. Daily cone beam computer tomography (CBCT) scans of the treated tumors were recorded on the linear accelerator to verify patient position. Total treatment time from first chemotherapy cycle to end radiotherapy was approximately 9.5 weeks.

Three patients were to be enrolled in part one. Eight blood samples per patient were drawn as follows. Two samples at baseline (test, retest) before administration of any therapy, one sample at 30 minutes, 1 and 2 hours after first infusion of cisplatin or carboplatin and one sample at 30 minutes, 1 and 2 hours after the first 2 Gy fraction on the first day of radiotherapy (Fig 1).

In part two of the study, six patients were enrolled with blood draws at 30 minutes post irradiation. AC and SCC were to be equally distributed with 3 AC and 3 SCC patients. Six blood samples were drawn during therapy; i.e. blood sample No. 1 was taken at baseline with no treatment given; sample No. 2 was drawn 1.5 weeks after the first cycle of chemotherapy at the time of a PET/CT scan for planning radiotherapy. Sample No. 3 was obtained on the first day of radiotherapy, and three further samples were taken every during radiotherapy: No. 4 after 11 fractions, No. 5 after 22 fractions, and No. 6 after 33[rd] and final fraction (Fig 1).

## Genomic testing

Blood samples were collected in 10 ml cell-free blood collection tubes (Streck®) or PAX tubes (Qiagen®), containing a preservative, which prevent the release of genomic DNA from blood cells, thus allowing isolation of cfDNA. Within 1–4 days after collection, plasma was isolated from the blood samples by centrifugation (2250 x g for 10 min), followed by a second centrifugation step (18000 x g for 10 min). Purification of cfDNA was done from 4 ml plasma using the QiaSymphony Circulating DNA kit (on the QiaSymphony, Qiagen). The cfDNA was eluted in a total volume of 60 µl. The cfDNA concentration was measured on the Qubit instrument (Thermo Fisher Scientific) using the double stranded DNA High Sensitivity Assay kit (Thermo Fisher Scientific). To confirm cfDNA counts to contain DNA from tumor cells, part two blood samples were analyzed using sWGS assessing somatic copy number alterations (SCNAs) in a genome-wide manner. Genomic alterations including SCNAs are a key characteristic of the cancer genome [15] and the detection of SCNAs in cfDNA can thus indicate the presence of tumor DNA in the circulation. The principle of sWGS relies on low-depth ($>$0.1x) whole-genome sequencing to estimate the SCNAs from the sequencing read depth [16], [17]. Indexed DNA libraries were constructed from 10 ng of cfDNA using the NEBNext Ultra II DNA library Prep Kit for Illumina (New England Biolabs) and paired-end sequenced (150-bp) on the MiSeq or NextSeq platforms (Illumina). Prior to sequencing, the libraries were

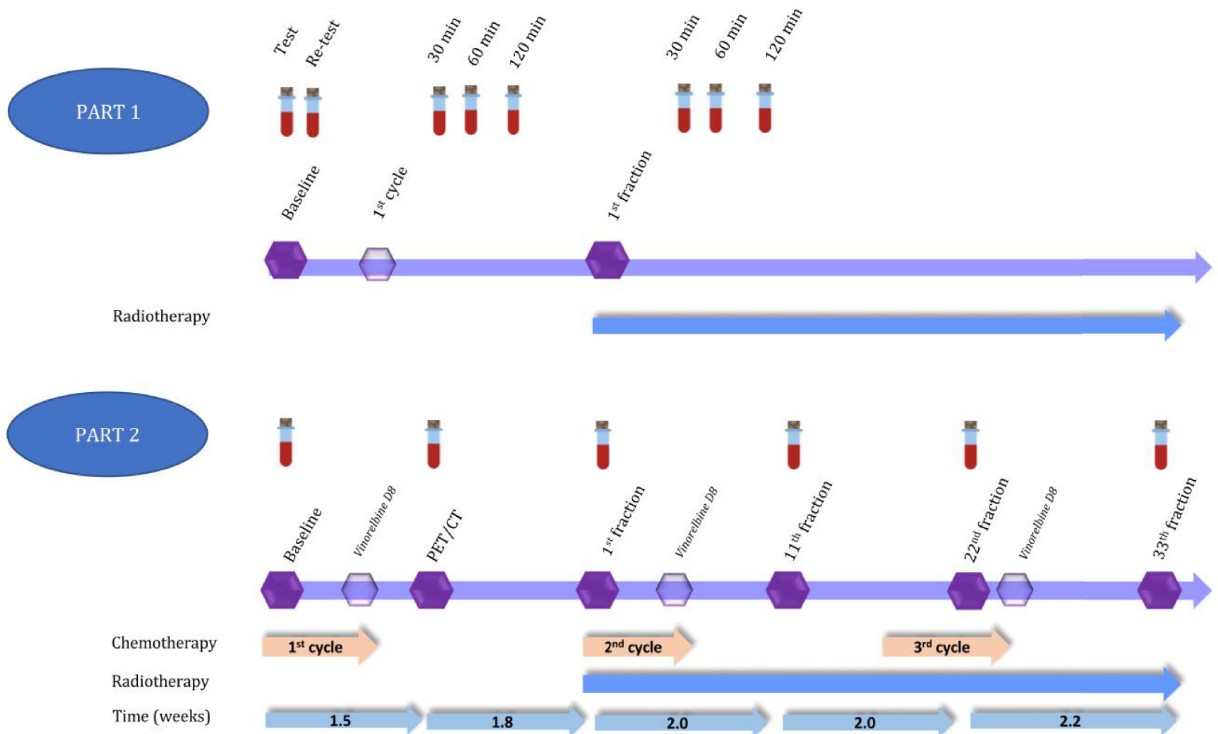

**Fig 1. First and second part of cfDNA study.** Part 1: Eight blood samples for cfDNA analysis taken in 4 patients at different timepoints during treatment. Part 2: Six blood samples for cfDNA analysis taken in 6 patients at different time-points during treatment. All samples in part 2 were taken approximately 30 minutes post-irradiation based on the results of part 1.

quantified using the high sensitivity dsDNA assay on the Qubit instrument (Thermo Fisher Scientific) and the fragment size inspected using the Bioanalyzer High Sensitivity DNA Analysis on the 2100 Bioanalyzer Instrument (Agilent). Furthermore, the libraries were pooled in equimolar amounts with six libraries in each run. A minimum of 5 million reads per sample were required to evaluate copy number alterations in cfDNA. Analysis of sWGS data was performed using the QDNASeq (https://bioconductor.org/packages/release/bioc/html/QDNAseq.html) dividing the genome into windows or bins of a certain size, and the GC-corrected read count for each bin is then used to determine if the region is increased or decreased.

Due to the low content of tumor cfDNA in this group of patients, an *in silico* size selection algorithm selecting fragments between 90 and 150 bp was applied to increase the detection of SCNAs [17]. This metric is called trimmed median absolute deviation from copy number neutrality (tMAD) and has been described by Mouliere and colleagues [17].

## Results

### Study cohort and feasibility of blood sampling

Ten patients were included in the study, see Table 1. Two patients withdrew from part one of the study after the first five samples due to the discomfort of blood sampling. Patient no 3 had a suboptimal blood sample for the baseline re-test and a cfDNA count could not be obtained. Consequently, an additional patient was included to obtain information on the optimal time for blood sampling after the first radiotherapy fraction. Sample two from patient no 8 did not

**Table 1. Patient, tumor and baseline cfDNA characteristics.**

| Patient no | Age | Gender | Histology | TNM | Clinical Stage | GTV cm3 | Volume cm3 receiving 59.4 Gy | Baseline cfDNA ng/ml |
|---|---|---|---|---|---|---|---|---|
| 1 | 69 | F | AC | T2N2 | IIIA | 40 | - | 5.85 |
| 2 | 69 | M | SCC | T2N2 | IIIA | 71 | - | 17.78 |
| 3 | 65 | F | SCC | T3N3 | IIIB | 144 | - | 1.50 |
| 4 | 68 | F | SCC | T2N3 | IIIB | 132 | - | 4.13 |
| 5 | 69 | F | AC | T4N3 | IIIB | 153 | 815 | 3.60 |
| 6 | 60 | F | AC | T1N3 | IIIB | 13 | 166 | 2.79 |
| 7 | 57 | M | SCC | T4N2 | IIIB | 290 | 1088 | 7.85 |
| 8 | 67 | M | SCC | T2N2 | IIIA | 158 | 659 | 8.27 |
| 9 | 69 | M | AC | T2N2 | IIIA | 78 | 422 | 2.52 |
| 10 | 81 | M | SCC | T4N1 | IIIA | 49 | 423 | 1.47 |

F: Female. M: Male. AC: Adenocarcinoma. SCC: Squamous cell carcinoma. TNM: Tumor Node Metastasis. All patients were M0. Clinical stage according to 7[th] version of UICC: Union for International Cancer Control. GTV: Gross tumor volume of tumor and lymph nodes. Volume cm3 receiving 90% of the prescribed radiation dose of 66Gy (59.4 Gy). Gy: Gray. -: not relevant. cfDNA: circulating cell free deoxyribonucleic acid. For patient 1–4 cfDNA counts are mean counts from the test and re-test at baseline.

reach the required number of reads (4 million reads), however could be assessed for further analysis.

Based on data from part one, a scatter plot was produced to identify any dynamics of cfDNA concentrations when obtaining cfDNA at different time intervals post chemotherapy/ radiotherapy. We saw no evidence of a relevant dynamics of cfDNA at this timescale (<2 hours) in either treatment modalities (Fig 2).

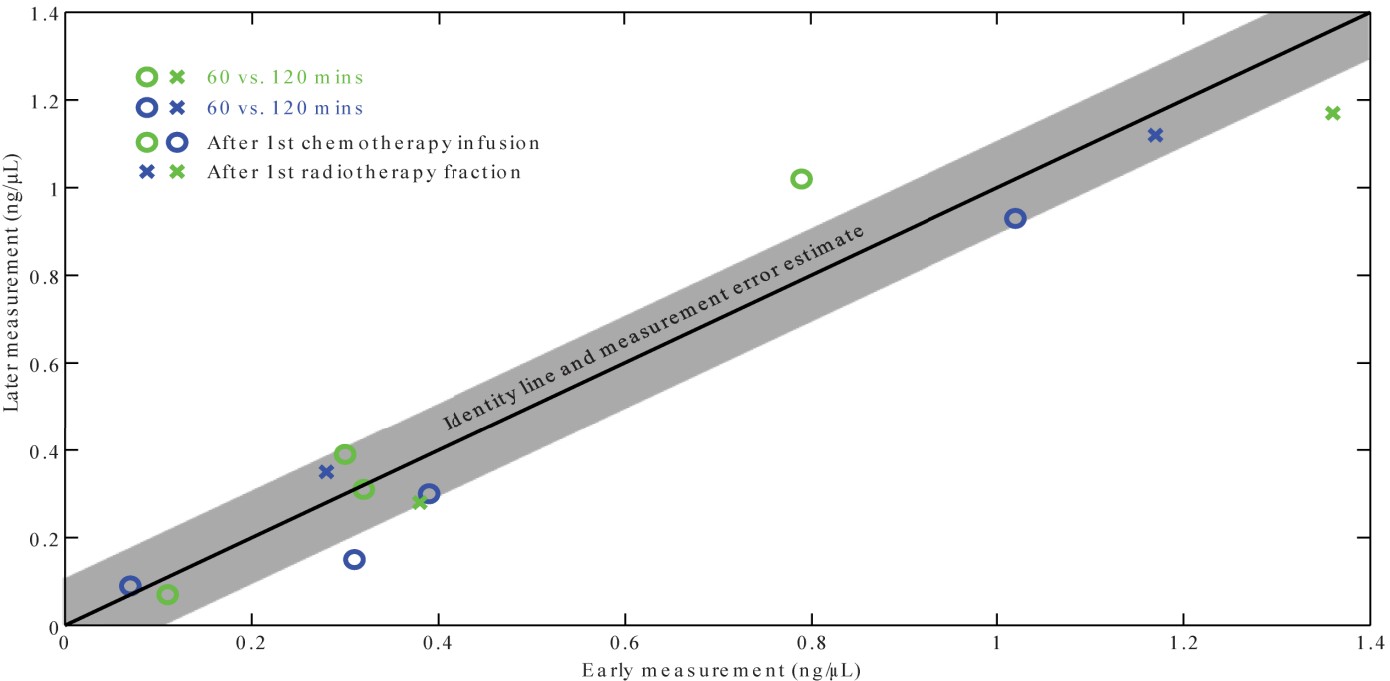

**Fig 2. Comparison of cfDNA concentrations at several timepoints after infusion of chemotherapy (circles) or first radiotherapy fraction (crosses).** Measurement uncertainty is estimated from test-retest at baseline.

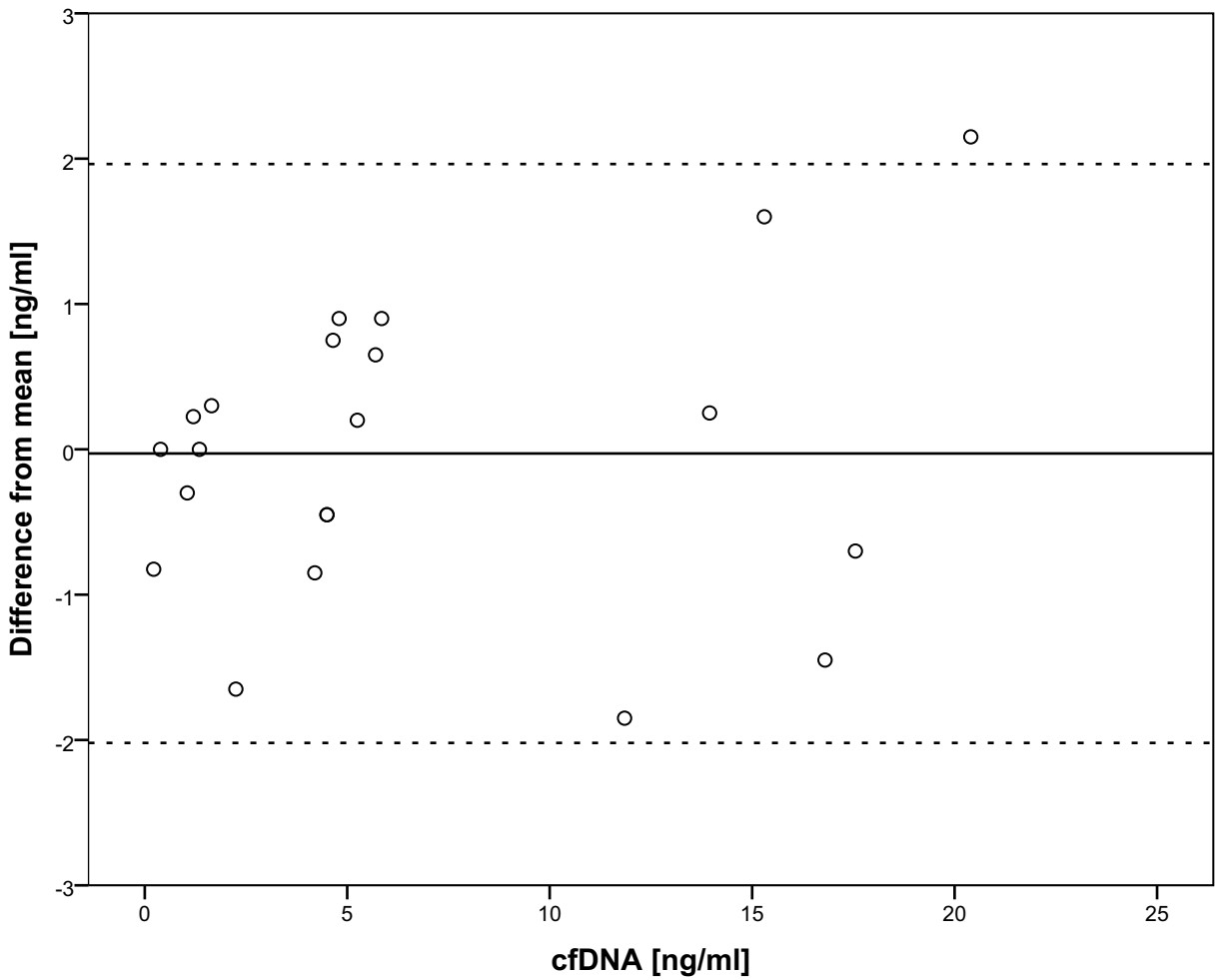

**Fig 3. Bland Altman plot of cfDNA concentrations shows detection with 95% tolerance limits of +/- 2 ng/ml cfDNA at baseline, and during the first 2 hours after treatment of either chemotherapy or radiotherapy.** Black horizontal line: Mean difference from a one sample T-test of all 25 samples. Dashed lines: limits of agreements (95%): mean difference +/- (SD*1.96). SD: standard deviation. cfDNA: circulating cell free DNA.

A Bland-Altman plot was produced of cfDNA measurements at all time points (baseline test, retest; after first chemotherapy infusion at 30, 60 and 120 minutes and after first radiotherapy fraction at 30, 60 and 120 minutes). For each sample, the difference from the mean of the samples acquired at baseline, after chemotherapy and after radiotherapy, respectively, is displayed (Fig 3). In the absence of any apparent time-dependence of the cfDNA counts, blood sampling for part two of the study was determined to be 30 minutes after chemo- or radiotherapy. The median cfDNA count at baseline in all ten patients was 3.86 ng/ml plasma, range [1.47–17.78].

## Response to radiation therapy and dynamics of cfDNA

Over the course of radiation therapy (from 1st to last 33rd fraction), tumor volume regression was observed in all 6 patients with a mean decrease of the maximum tumor diameter of 16%, range [1–24%] measured in one direction of the primary tumor on the CBCT scans (Fig 4, top row).

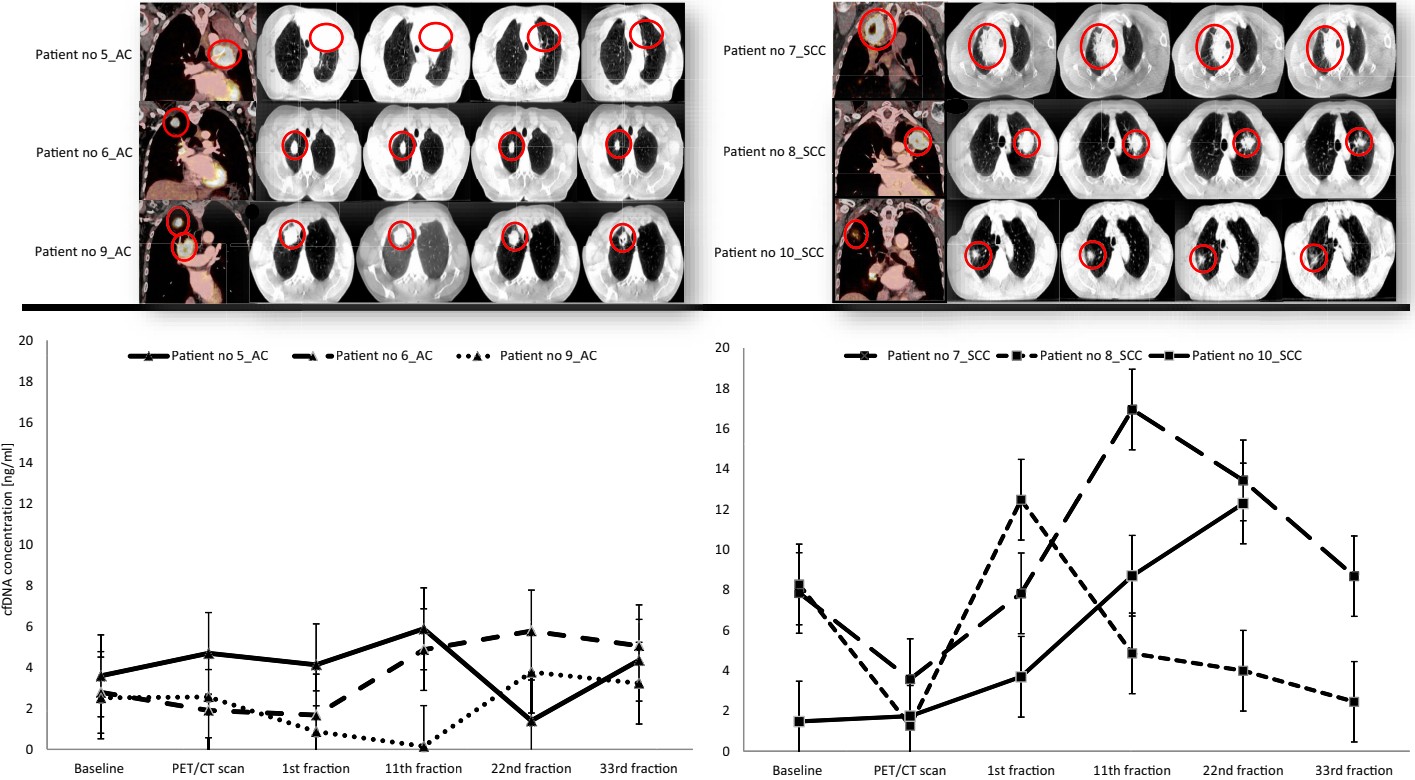

**Fig 4. PET/CT, CBCT scans and cfDNA concentrations from baseline and during radiation therapy in adenocarcinoma (AC) patients and patients with Squamous Cell Carcinoma (SCC) from part 2 of the study.** Error bars show +/- 2 ng/ml cfDNA for each sample. Notice: Patient no 10_SCC missed the blood sample at 33rd fraction. CBCT: cone beam computer tomography.

When looking at the dynamics of cfDNA, AC and SCC patients displayed different patterns. AC patients had lower mean cfDNA concentration at baseline than SCC, 2.97 and 5.86 ng/ml plasma, respectively. Furthermore, fluctuations in cfDNA during therapy were less pronounced in AC patients compared to SCC patients. For example, the mean increase in cfDNA from baseline to the 22nd fraction (blood sample no 5) was 0.69 ng/ml range [-2.2 to 3.0] ng/ml for the three AC patients and 4.0 ng/ml range [-4.3 to 10.8] ng/ml in the three SCC patients (Fig 4). This difference was not statistically significant, however (Wilcoxon rank sum test; p-value = 0.7). Noticeably, we observed moderate–if any—correlation between total cfDNA and tumor shrinkage, Spearman's rho of 0.54 at 11th fraction and 0.77 at 22nd fraction (S1 Fig).

**Dynamics of ctDNA during therapy.** To estimate the proportion of cfDNA originating from tumor (ctDNA), we further assessed the presence of somatic copy number alterations (SCNAs) using sWGS in 6 patients (Fig 5). Blood samples were obtained at baseline, at the time of the FDG PET/CT scan for radiotherapy planning, and after the 1st and 11th RT fraction; i.e. 4 samples per patient. However, only two samples from patient 9 were available for analysis. An additional sample from patient 10 (after 22nd fraction) was included.

In general, samples showed silent profiles across the two cancer types; i.e. no SCNAs were detected most probably due to the low content of ctDNA in the plasma (Fig 5). However, patient 7 had segmental amplification on chromosome 7 involving *EGFR*. The amplification was detected at baseline and in the following two blood samples. Furthermore, SCNAs were detected in patient 5 (gain 8q) and patient 9 (gain 5p) after the first fraction of radiotherapy.

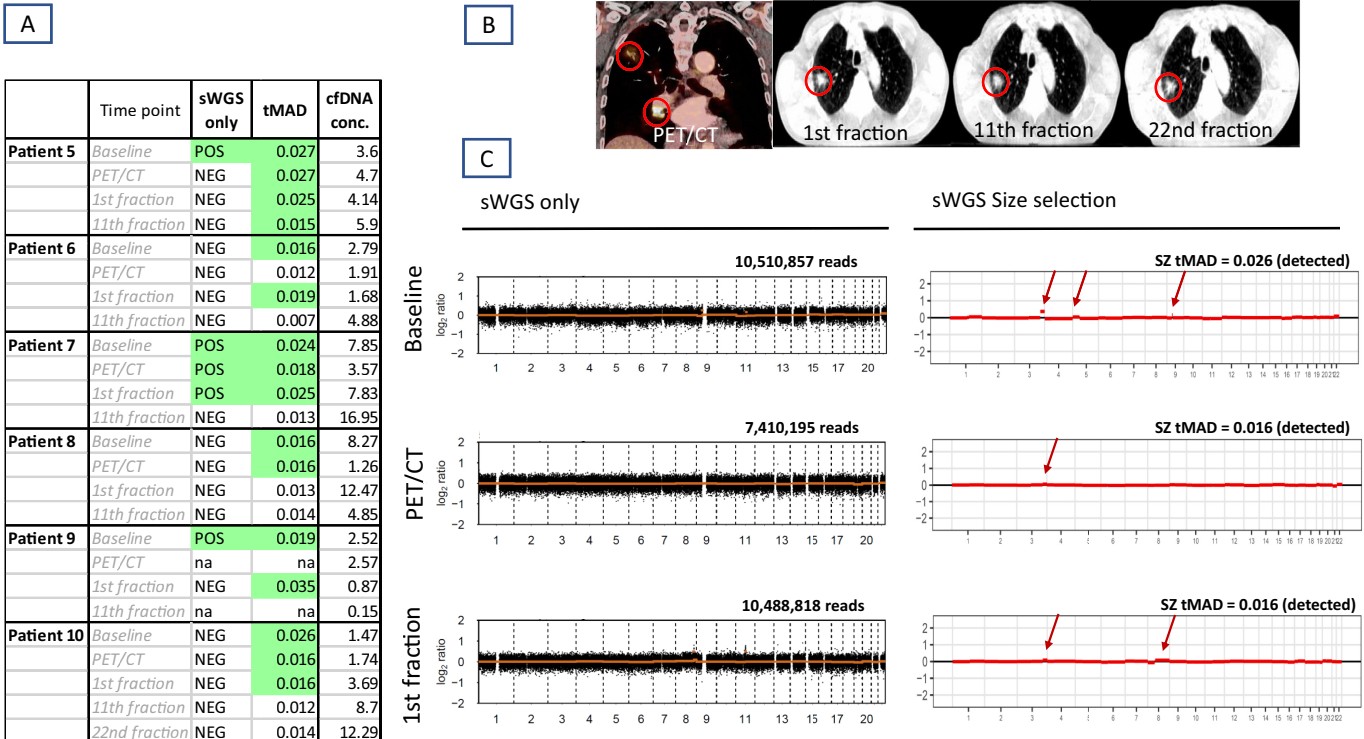

**Fig 5. Efficacy of sWGS and subsequent in silico size selection for detection of Somatic Copy Number Alterations (SCNA) in AC and SCC patients.** (A) Table showing overview of SCNA detected by sWGS standard processing (sWGS only) and increase of detection sensitivity after application of size selection (tMAD). (B) Patient no 10 at PET/CT scan prior to radiotherapy, and CBCT scans at the 1st, 11th and 22nd radiation fraction. (C) Plots from Patient no 10. Left: Plots of log2 ratios after standard sWGS processing (only silent profiles). Right: after the size selection for cfDNA collected at baseline, PET CT scan and at the first radiation fraction. x-axis indicates the chromosomes and y-axis log2 ratios. Red arrows indicate copy number alterations.

S2 Fig depicts all sWGS reads and corresponding tMAD scores in patients from part two of the study.

As cfDNA from tumor cells is shorter than cfDNA originating from normal cells [18],[19], we further applied *in silico* size selection in order to increase the sensitivity of SCNAs detection. Using size selection and tMAD scores have shown ctDNA detection down to a minor allele frequency of 0.01 (1%) [17,20]. Noticeably, processing sWGS data by size selection with tMAD > 0.015, SCNAs were detected in 16 out of 23 samples (70%) compared to 5 samples (22%) without size selection (Fig 5). For example, patient 10 showed silent profiles in all samples by basic sWGS processing. After size selection, SCNAs were found on chromosome 3, 5 and 9 at baseline. After the first round of chemotherapy, i.e. at PET/CT scan, only gain 3q was present. However, already after the first fraction of radiotherapy, additional aberration on 8q was revealed, involving *MYC* amplification (Fig 5). This could indicate possible activation of a resistance mechanism [21].

Of note, the increase in cfDNA concentration did not reflect the increase in ctDNA content as indicated by presence of SCNAs. To further examine the reason for increase of normal cell death, we looked on possible inflammation and/or normal tissue necrosis caused by radiation. However, leucocytes and platelets tended to decrease during therapy. Leucocytes decreased on average 5.3 units (std. dev. 1.6) from baseline (at mean $9.3 * 10^9/l$) to measurement between cfDNA samples four and five (11th to 22nd fraction), p<0.001 for difference. Platelets on average dropped by 147 units (std. dev. 98 units) from mean 424 ($10^9/l$) at baseline with p = 0.014

for difference in the same time span. For lactate dehydrogenase (LDH), baseline mean was 201 (U/l) and all measured stayed within normal range with p = 0.8 for difference over the time span. There were no obvious patterns in cfDNA changes during therapy on one hand and the volume of the body being irradiated or the size of the GTV on the other (S3 and S4 Figs).

**Patient treatment outcome.** Patient no 5 with AC was diagnosed with multiple brain metastases during radiotherapy, at the time around 22nd fraction (blood sample 5). She completed cCRT and was given whole brain radiation hereafter. She died six months post therapy. Patient no 6 was offered surgery after cCRT. Histology showed active adenocarcinoma in the primary tumor and affected lymph node. At the latest follow up seven months after, she was tumor free assessed from a PET/CT scan. Patient no 9 also received salvage surgery after completion of cCRT. Histology showed active AC in the right upper lobe. Lymph node stations 4 and 10 were without tumor cells after cCRT. He was without any evidence of disease six months post cCRT and will be offered three additional cycles of chemotherapy. Patient no 7 had loco-regional relapse two months after cCRT. Patient no 8 had surgery post cCRT. Histology from the operated lobe showed active squamous cell carcinoma in the primary tumor. Lymph nodes were without cancer cells. He was without any evidence of disease eight months post cCRT. Patient no 10 completed cCRT but died ten days later. In conclusion, after completion of cCRT, five out of six patients had active disease, either by clinical and radiological progression or pathological examination at surgery.

## Discussion

This pilot study shows that cfDNA concentrations in patients undergoing cCRT can be assessed with stable cfDNA measurements from 30 to 120 minutes after a radiation therapy fraction. Concentrations of cfDNA during treatment showed mean increases above the 95% tolerance limits of +/- 2 ng/ml cfDNA established in part 1 in SCC patients, whereas cfDNA changes in AC stayed within the tolerance limits. sWGS analyses could *not* confirm that observed cfDNA changes represent changes in ctDNA content. In most cfDNA samples, no genomic alterations were detected after the 11th fraction. However, size selection with tMAD scores found surprisingly more ctDNA in most samples. It further confirmed the sWGS results, that ctDNA was not the cause of high cfDNA levels past the 11th fraction of radiotherapy, as the tMAD scores were below 0.015 for the remaining course of radiotherapy. However, patient cases illustrate that for most tumors, ongoing activity was observed in five out of six patients post cCRT. A silent ctDNA profile did not seem to correspond to clinical outcome of the patients. Likewise, there was no concordance between radiological response and cfDNA change during treatment. Generally, the volume of the primary tumor on CBCT decreased but the cfDNA counts increased during therapy in both SCC and AC patients. Interestingly, patient 5 had low cfDNA counts with only one sample showing a chromosome 8 amplification during therapy. She had a marked decrease in chest tumor size, yet multiple brain metastases were discovered around the time of the 22nd fraction.

How radiotherapy influences the tumor cells and surrounding normal tissue to release DNA remains to be clarified. One hypothesis is that cfDNA counts increase as more and more tumor cells are exposed to ionizing radiation hereby releasing DNA into the blood stream. A quantitative assessment of tumor DNA as a proportion of cfDNA counts is currently under investigation as the sensitivity of sWGS in detecting ctDNA among high cfDNA levels is still not clear. This pilot study was not powered to detect a pre-specified difference in the cfDNA profile in SCC versus AC, nor to detect a possible correlation between cfDNA differences in AC and SCC, but the observed patterns are interesting for future studies and suggest that the two histologies should be separated in future analyses.

In conclusion, future studies of cfDNA could conveniently be conducted with blood samples obtained approximately 30 minutes post chemo- and radiotherapy treatment, and moderate variation in sample timing does not appear to affect results. cfDNA concentrations change during therapy beyond the sample-to-sample variability. However, more studies are needed to clarify if these changes are of value in guiding further therapy and to understand the mechanism behind cfDNA release during therapy.

## Supporting information

**S1 Fig. Scatter plot of cfDNA change as a function of tumor shrinkage measured on CT.** •: Tumor change from PET/CT scan (blood sample no 2) to fourth cfDNA blood sample (11[th] fraction). +: Tumor change from PET/CT scan (blood sample no 2) to fifth cfDNA blood sample (22[nd] fraction). cfDNA: circulating cell-free DNA.
(PDF)

**S2 Fig. All sWGS reads following the in silico size selection data with tMAD scores in patients from part 2 of the study.**
(PDF)

**S3 Fig. Scatter plot of volume [cm3] of the body receiving 59.4 Gy (90% of the prescribed 66 Gy) and a function of cfDNA change.** Gy: Gray. cfDNA: circulating cell free DNA. PET/CT: positron emission tomography/computer tomography.
(PDF)

**S4 Fig. Scatter plot of cfDNA level change as a function of GTV divided by histology.** cfDNA: circulating cell free DNA. GTV: Gross tumor volume. PET/CT: positron emission tomography/computer tomography. SCC: Squamous cell carcinoma. AC: Adenocarcinoma.
(PDF)

## Author Contributions

**Conceptualization:** Lotte Nygård, Gitte F. Persson, Barbara M. Fischer, Seppo W. Langer, Andreas Kjær, Ivan R. Vogelius, Søren M. Bentzen.

**Data curation:** Lotte Nygård, Lise B. Ahlborn, Dineika Chandrananda, Miglė Gabrielaite, Nitzan Rosenfeld, Florent Mouliere, Olga Østrup, Ivan R. Vogelius, Søren M. Bentzen.

**Formal analysis:** Lotte Nygård, Gitte F. Persson, Ivan R. Vogelius, Søren M. Bentzen.

**Funding acquisition:** Ivan R. Vogelius, Søren M. Bentzen.

**Investigation:** Lotte Nygård, Jonathan W. Langer, Ivan R. Vogelius, Søren M. Bentzen.

**Methodology:** Lotte Nygård, Gitte F. Persson, Seppo W. Langer, Nitzan Rosenfeld, Florent Mouliere, Olga Østrup, Ivan R. Vogelius, Søren M. Bentzen.

**Project administration:** Lotte Nygård.

**Supervision:** Gitte F. Persson, Seppo W. Langer, Andreas Kjær, Olga Østrup, Ivan R. Vogelius, Søren M. Bentzen.

**Writing – original draft:** Lotte Nygård.

**Writing – review & editing:** Lise B. Ahlborn, Gitte F. Persson, Dineika Chandrananda, Jonathan W. Langer, Barbara M. Fischer, Seppo W. Langer, Miglė Gabrielaite, Andreas Kjær, Nitzan Rosenfeld, Florent Mouliere, Olga Østrup, Ivan R. Vogelius, Søren M. Bentzen.

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
