## [Decision Letter · Decision Letter 0]

9 Jan 2020

PONE-D-19-24047

Circulating cell free DNA during definitive chemo-radiotherapy in non-small cell lung cancer patients – initial observations

PLOS ONE

Dear Dr. Nygård,

Thank you for submitting your manuscript to PLOS ONE. After careful consideration, we feel that it has merit but does not fully meet PLOS ONE’s publication criteria as it currently stands. Therefore, we invite you to submit a revised version of the manuscript that addresses the points raised during the review process.

Please address the single reviewer's comments.

To enhance the reproducibility of your results, we recommend that if applicable you deposit your laboratory protocols in protocols.io, where a protocol can be assigned its own identifier (DOI) such that it can be cited independently in the future. For instructions see: http://journals.plos.org/plosone/s/submission-guidelines#loc-laboratory-protocols

We look forward to receiving your revised manuscript.

Kind regards,

Jeffrey Chalmers, Ph.D.

Academic Editor

PLOS ONE

Journal Requirements:

2. In your Methods section, please provide additional information about the participant recruitment method. Specifically, please provide the dates of enrolment (month/year) and a description of how the participants were recruited.

3. Your ethics statement must appear in the Methods section of your manuscript. If your ethics statement is written in any section besides the Methods, please move it to the Methods section and delete it from any other section. Please also ensure that your ethics statement is included in your manuscript, as the ethics section of your online submission will not be published alongside your manuscript.

Reviewers' comments:

Reviewer's Responses to Questions

**Comments to the Author**

1. Is the manuscript technically sound, and do the data support the conclusions?

Reviewer #1: Yes

2. Has the statistical analysis been performed appropriately and rigorously? 

Reviewer #1: Yes

3. Have the authors made all data underlying the findings in their manuscript fully available?

Reviewer #1: Yes

4. Is the manuscript presented in an intelligible fashion and written in standard English?

Reviewer #1: Yes

5. Review Comments to the Author

Reviewer #1: • The data demonstrate that there was no concordance between radiological response and cfDNA change during treatment. This in line with previous observations which have shown a lack of predictive accuracy for total cfDNA (Weiss G et al, Clin Cancer Res 2017; 23: 5074-81). The paper of Weiss, however, demonstrated that on the other hand chromosomal instability quantification in plasma cfDNA was associated with therapeutic response to immune therapy. This paper should be mentioned in the publication.

• P. 6: The authors state that samples were analyzed “… using sWGS, resembling the method of plasma-Seq” and give a citation. Some more detailed information on how they applied this method should perhaps be given in the supplement.

• To what extent was the sensitivity of somatic copy number alterations detection increased?

• P. 9: The authors state that the mean increase in cfDNA was different in AC and SCC patients. Was this difference statistically significant?

• P. 11: What was the detectable ctDNA fraction with their method? In this context, the authors could cite a paper of Fiala et al (J Appl Lab Med 2018; 03: 300-313) which provides information on the likelihood of ctDNA detection related to detectable fraction.

6. PLOS authors have the option to publish the peer review history of their article (what does this mean?). If published, this will include your full peer review and any attached files.

Reviewer #1: Yes: Michael Oellerich

---

## [Author Response · Author response to Decision Letter 0]

28 Feb 2020

Thank you very much for the revisions of our manuscript. The comments have been addressed in "Response to reviewers"

---

## [Editor Report · Decision Letter 1]

3 Apr 2020

Circulating cell free DNA during definitive chemo-radiotherapy in non-small cell lung cancer patients – initial observations

PONE-D-19-24047R1

Dear Dr. Nygård,

We are pleased to inform you that your manuscript has been judged scientifically suitable for publication and will be formally accepted for publication once it complies with all outstanding technical requirements.

With kind regards,

Jeffrey Chalmers, Ph.D.

Academic Editor

PLOS ONE
---

## [Editor Report · Acceptance letter]

8 Apr 2020

PONE-D-19-24047R1 

Circulating cell free DNA during definitive chemo-radiotherapy in non-small cell lung cancer patients – initial observations 

Dear Dr. Nygård:

I am pleased to inform you that your manuscript has been deemed suitable for publication in PLOS ONE. Congratulations! Your manuscript is now with our production department. 

With kind regards,

on behalf of

Dr. Jeffrey Chalmers 

Academic Editor

PLOS ONE